# Distinct functional elements for outer-surface anti-interference and inner-wall ion gating of nanochannels

Pengcheng Gao[1], Qun Ma[1], Defang Ding[1], Dagui Wang[1], Xiaoding Lou[1], Tianyou Zhai [2] & Fan Xia[1]

Over the decades, widespread advances have been achieved on nanochannels, including nanochannel-based DNA sequencing, single-molecule detection, smart sensors, and energy transfer and storage. However, most interest has been focused on the contribution from the functional elements (FEs) at the inner wall (IW) of nanochannels, whereas little attention has been paid to the contribution from the FEs at the nanochannels' outer surface (OS). Herein, we achieve explicit partition of $FE_{OS}$ and $FE_{IW}$ based on accurate regional-modification of OS and IW. The $FE_{IW}$ are served for ionic gating, and the chosen $FE_{OS}$ (hydrophobic or charged) are served for blocking interference molecules into the nanochannels, decreasing the false signals for the ionic gating in complex environments. Furthermore, we define a composite factor, areas of a radar map, to evaluate the $FE_{OS}$ performance for blocking interference molecules.

[1] Engineering Research Center of Nano-Geomaterials of Ministry of Education, Faculty of Materials Science and Chemistry, China University of Geosciences (CUG), 388 lumo Road, 430074 Wuhan, China. [2] State Key Laboratory of Material Processing and Die & Mould Technology, School of Material Sciences and Engineering, Huazhong University of Science and Technology (HUST), 430074 Wuhan, China. These authors contributed equally: Pengcheng Gao, Qun Ma. Correspondence and requests for materials should be addressed to F.X. (email: xiafan@cug.edu.cn)

Biological channel proteins, embedded in lipid bilayer, act as nanochannels regulating the transmission of various biomolecules and ions, which is vital to life process[1,2]. Scientists modify the nanochannels, obtained by extraction or biomimetic synthesis, with function elements (FEs) for the manipulation of ion translocation[3–10]. During the decades, widespread advances have been achieved on nanochannels correspondingly, including nanochannel based DNA sequencing[11–14], single-molecular detections[15–18], smart sensors[19–22], energy transfer/storage[23–25], and so on[26–30]. However, researchers focus all interests on the contribution from the FEs at the inner wall (IW) of nanochannels ($FE_{IW}$), little attention has been paid on the contribution from the FEs at the outer surface (OS) of nanochannels ($FE_{OS}$)[10,31–33]. Currently, theory simulations have forecasted the respective contribution from the FEs at OS and IW to ionic gating[34,35]. However, till now, experimental demonstration is rare. An essential issue has been proposed: what roles do the $FE_{OS}$ play in nanochannels? If we failure to settle this issue, it will be an obstacle to the nanochannel-based applications such as DNA sequencing[36–38], molecule sensing[39–43], and so on[24–26]. For instance, recognition a nucleotide base in sequencing typically relies on a subtle distinction of current and dwell time signatures by the blockade of inner nanochannels, which, however, cannot be actualized when the base has not access to the inner of nanochannel[44]. On the other hand, it thus may be envisioned that elaborate $FE_{OS}$ would be probably beneficial to differentiate a specific base in a DNA sequence.

In a very recent work, we made a preliminary experimental to distinguish the contribution of $FE_{IW}$ and $FE_{OS}$ to ionic gating[45]. The $FE_{OS}$ were proved to synergistically enhance the ionic gating efficiency. However, the explicit partition of regional functionalization of $FE_{IW}$ and $FE_{OS}$ has not been demonstrated. Furthermore, the independent function of the $FE_{OS}$ has not been achieved.

Additionally, the nanochannel, could also be nonspecifically blocked by interfering molecules, producing even false signals especially in a complex system, which would severely reduce the performance for DNA sequencing, molecule sensing and et al.[46,47] We expect that the rational $FE_{OS}$ would block the interference molecules to reduce the false signal. Furthermore, it is expected that distinct $FE_{OS}$ and $FE_{IW}$ would invest a

nanochannel with distinct but coordinate functions to accomplish the above tasks in complex environments, which only could be achieved by the functions of $FE_{OS}$ and $FE_{IW}$ rather than sole $FE_{IW}$. Unfortunately, to date, all these concerns remain untapped.

Herein, we achieve the explicit partition of $FE_{OS}$ and $FE_{IW}$ based on the accurate regional-modification of OS and IW. The $FE_{IW}$ are served for ionic gating, and the chosen $FE_{OS}$ (hydrophobic or charged) are served for blocking the interference molecules into the nanochannels, decreasing the false signals for the ionic gating in complex environments (Fig. 1). Furthermore, we also define a composite factor, areas of a radar map, to evaluate the $FE_{OS}$ performance for blocking interference molecules.

## Results

**Explicit spatial partition of OS and IW.** To create distinct OS and IW, we deposited ITO and Au alternately at the one side of the AAO membrane's OS (Supplementary Fig. 1)[48]. The deposited speed is as low as 0.01 nm s$^{-1}$. The linear scanning of energy dispersive spectrometer along the z-axil direction of nanochannels demonstrates that the elements as Au and Indium exist at the tips of a nanochannel (Supplementary Fig. 2). We designed four stacking orders of Au and ITO, on purpose of creating four modes of Au distributions as shown in Fig. 2a–m (none means that there is no FEs at the specific region, and all names of used samples are listed in Supplementary Table 1 and all abbreviations are listed in Supplementary Table 2).

Accurate characterizations of elements along nanochannels have been done by a time of flight secondary ion mass spectrometry (TOF-SIMS). The analyzing area is around 60 μm × 60 μm on the X–Y plane beginning from the outmost membranes. The deposition depths (sectional-view of SEM images in Supplementary Fig. 3) of Au and ITO along nanochannels are characterized by the intensity distribution vs. depth from TOF-SIMS. We define the boundary at 5% of the peak intensity (Fig. 2b–n) as the start or the end of Au or ITO deposition. Corresponding 3D reconstructions are shown in Fig. 2c–o. Importantly, the depth of Au or ITO on IW can reach several micrometers, in contrast with the several nanometers at OS. For example, for the 500-s Au deposition, the depth of Au can reach 1.37 μm on the IW, while the thickness of Au is

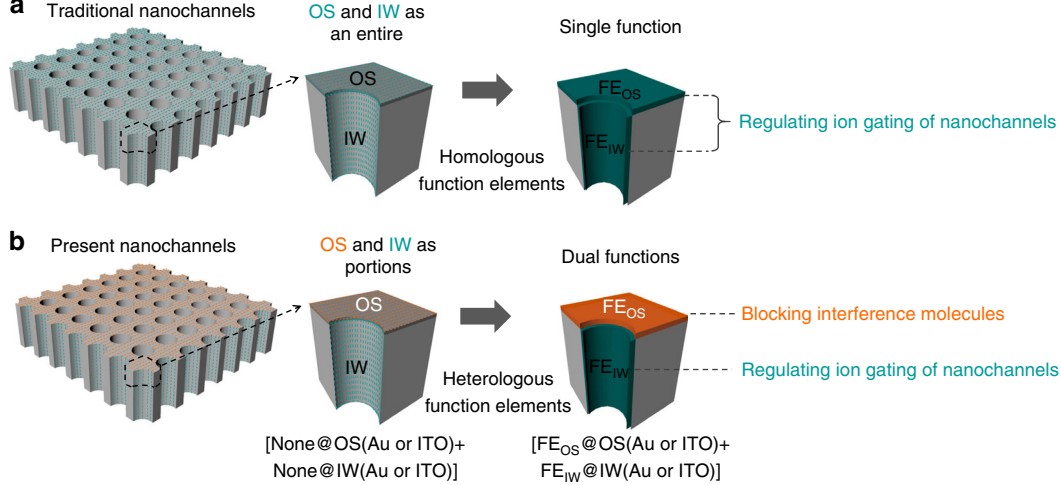

**Fig. 1** Nanochannels with dual functions. Traditional nanochannels with single function, while ours with dual functions. **a** $FE_{IW}$ at IW are considered as the main contributions to ion gating, while $FE_{OS}$ at OS are nearly ignored in traditional nanochannels. **b** The explicit partition of $FE_{OS}$ and $FE_{IW}$ are achieved based on the accurate regional-modification of OS and IW. More importantly, $FE_{IW}$ are served for ionic gating, and the chosen $FE_{OS}$ (hydrophobic or charged), to our knowledge, are served for blocking the interference molecules into the nanochannels, decreasing the false signals for the ionic gating, even in complex environments

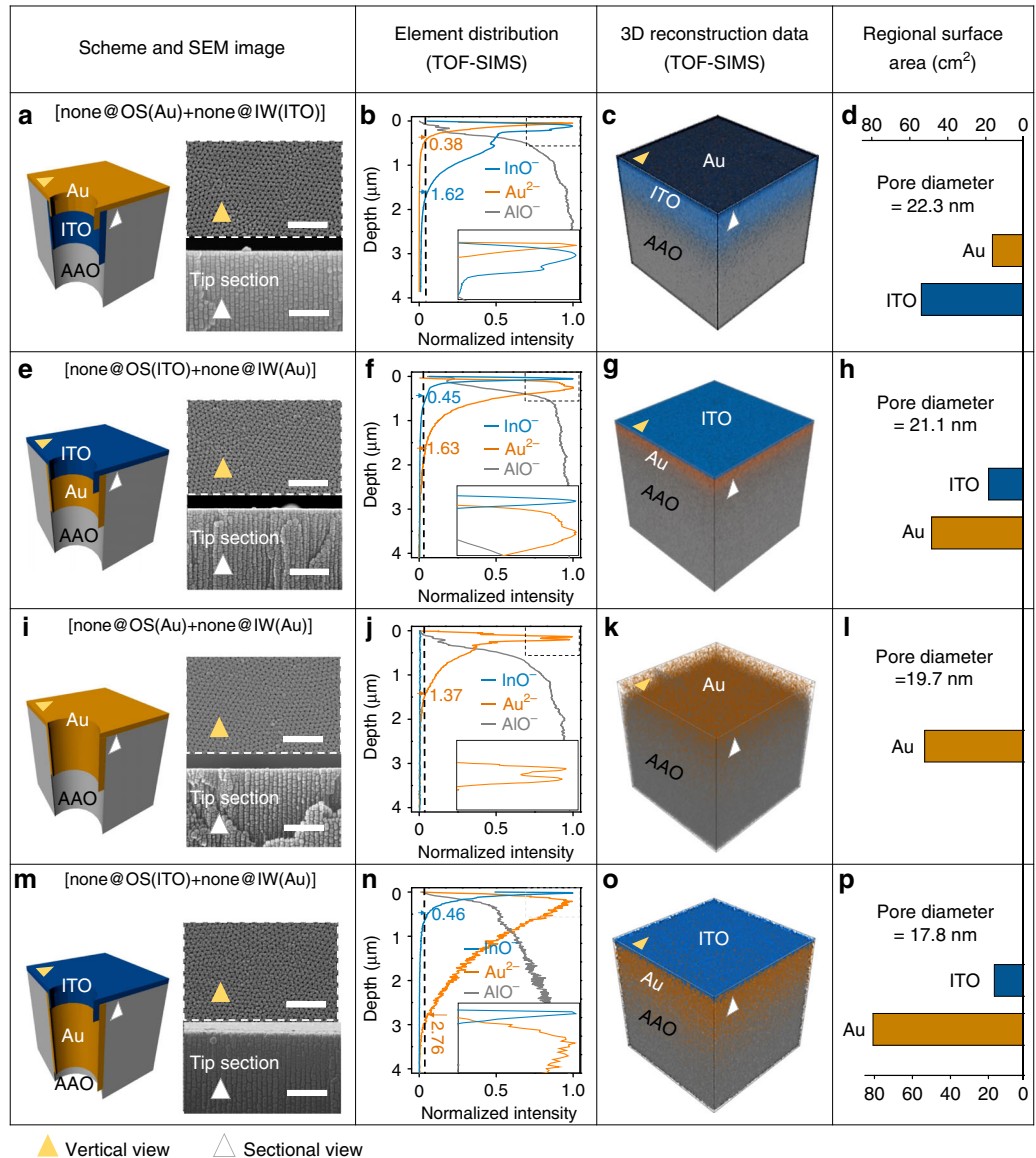

**Fig. 2** Explicit spatial partition of OS and IW. We design four samples with varied stacking orders of Au and ITO as **a** [none@OS(Au) + none@IW(ITO)], **e** [none@OS(ITO) + none@IW(Au)], **i** [none@OS(Au) + none@IW(Au)] and **m** [none@OS(ITO) + none@IW(Au)] by atomic layer deposition (ALD). Corresponding SEM images taken from vertical and sectional view reveal that nanochannels remain open even after the deposition processes. The element distributions of the four samples are characterized by TOF-SIMS in **b**, **f**, **j**, and **n**, respectively. The depth with the 5% of the peak intensity is defined as the start or the end of Au or ITO deposition (the gray dashed line in **b**, **f**, **j**, and **n**, respectively). Thus, the corresponding 3D reconstructions (**c**, **g**, **k**, and **o**) of the four samples are plotted from the intensity distribution of TOF-SIMS data. The surface area of Au and ITO on the OS or IW are calculated (**d**, **h**, **l**, and **p**) according to the element depth measured from TOF-SIMS and the pore size measured from SEM (Supplementary Fig. 5). Scale bars, 500 nm

just 5 nm on the OS (Supplementary Fig. 4). Furthermore, after the depositions, the tips of channels remain open as shown from not only the top view but also the sectional view (Right in Fig. 2a–m). Statistics of pore sizes have been done by counting around 9000 pores for each kind of deposited membrane. The average pore size decreases slightly from the original AAO with 25 nm diameter to 22.3 nm (Fig. 2a), 21.1 nm (Fig. 2e), 19.7 nm (Fig. 2i), 17.8 nm (Fig. 2m), respectively (Supplementary Fig. 5). And accurate area of OS and IW are listed in four samples (Fig. 2d–p). From the above, explicit spatial partition of OS and IW is realized with the control accuracy of dozens of nanometers.

**Explicit regional partition of FE_IW and FE_OS.** Since explicit spatial partition of OS and IW has been accomplished, accurate partition of $FE_{IW}$ and $FE_{OS}$ with different functional groups (–SH,

–OH, et al.) can be realized next. DNA oligomers are chosen as the FEs for regional modification and further ionic gating due to the facility for labeling and design of programmable structure (Fig. 3a–n). In detail, Au is modified with the thiol-DNA through Au-thiol binding, while ITO and AAO with hydroxyl groups are modified with the amino-DNAs through silane coupling reactions (Supplementary Fig. 6). We designed a linear 27 base DNA sequence with fluoresceine isthiocyanate (FITC) and Cyaine-5 (Cy5) respectively, used in our previous works (all sequences listed in Supplementary Table 2)[49]. Laser scanning confocal microscopes (LSCM) were applied to characterize the distribution of FEs at the OS and IW respectively. As shown in the LSCM images (Fig. 3d–p), the fluorescent distributions along nano-channels are divided into two or three regions, which are in line with the distribution of Au, ITO, and AAO. However, compared

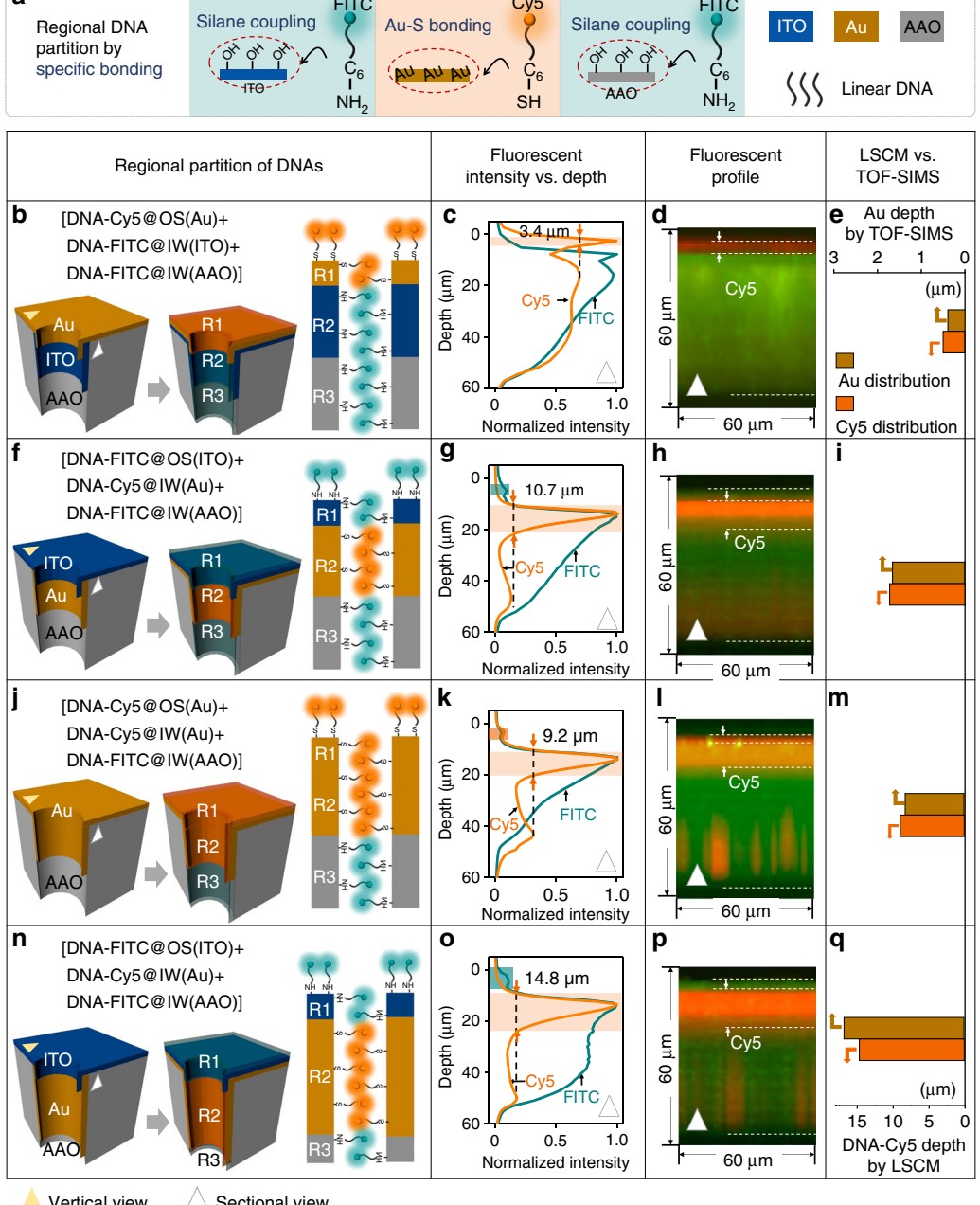

**Fig. 3** Explicit partition of FE$_{IW}$ and FE$_{OS}$. **a** DNA labeled with FITC or Cy5 are bonded at ITO, Au and AAO surface using silane coupling, Au-S bonding and silane coupling, respectively. Schematics (**b**, **f**, **j**, **n**) described the varied modification labeled DNA at the OS and IW of four samples which are introduced in Fig. 2a–m, respectively. The intensity distributions of fluorescence along the depth of AAO membrane are listed in **c**, **g**, **k**, **o**, respectively. The red lines and green lines represent the DNA with Cy5 and the DNA with FITC, respectively, based on the Z-axil of LSCMs for the four samples (**d**, **h**, **l**, **p**). **e**, **i**, **m**, **q** Comparison between the Au depth (gold) measured by TOF-SIMS and the DNA-Cy5 (red) depth measured by LSCM

with TOF-SIMS, the fluorescent distribution of DNA-Cy5 (Fig. 3c–o) is far greater than the depth of the deposited Au (Fig. 2a–m). The large disparity is attributed to the following reasons: (1) There are great disparities for the measuring accuracy between TOF-SIMS (about 10 nm) and LSCM (about 500 nm); (2) The measurement depth of Au from fluorescence is larger than the actual value; (3) The measurement depth of Au from TOF-SIMS is less than the actual value (see the detail discussion in the 4th part in Supplementary Information). Nevertheless, both results from TOF-SIMS and LSCM show the same trend for the depth of Au (Figs. 2d–p, 3e–q), which means that the data from the two instruments support each other. FE$_{OS}$ and FE$_{IW}$,

therefore, could be rationally coated on the desired OS and IW, respectively.

**Contribution from FE$_{OS}$ and FE$_{IW}$ to ionic gating**. The contributions from FE$_{OS}$ and FE$_{IW}$, respectively, to ionic gating were investigated, referring to the increment of gating ratio ($\Delta gr$) after modifying IW and OS with DNAs, respectively (Fig. 4b, c). The four patterns of the DNA-Cy5 distributions were designed for electrochemical characterizations in Fig. 4d–s. The fluorescent intensity referring to the amount of DNA-Cy5 was characterized by LSCM (Supplementary Fig. 7). A two-electrode system, previously reported[49,50], contained two symmetric Ag/AgCl

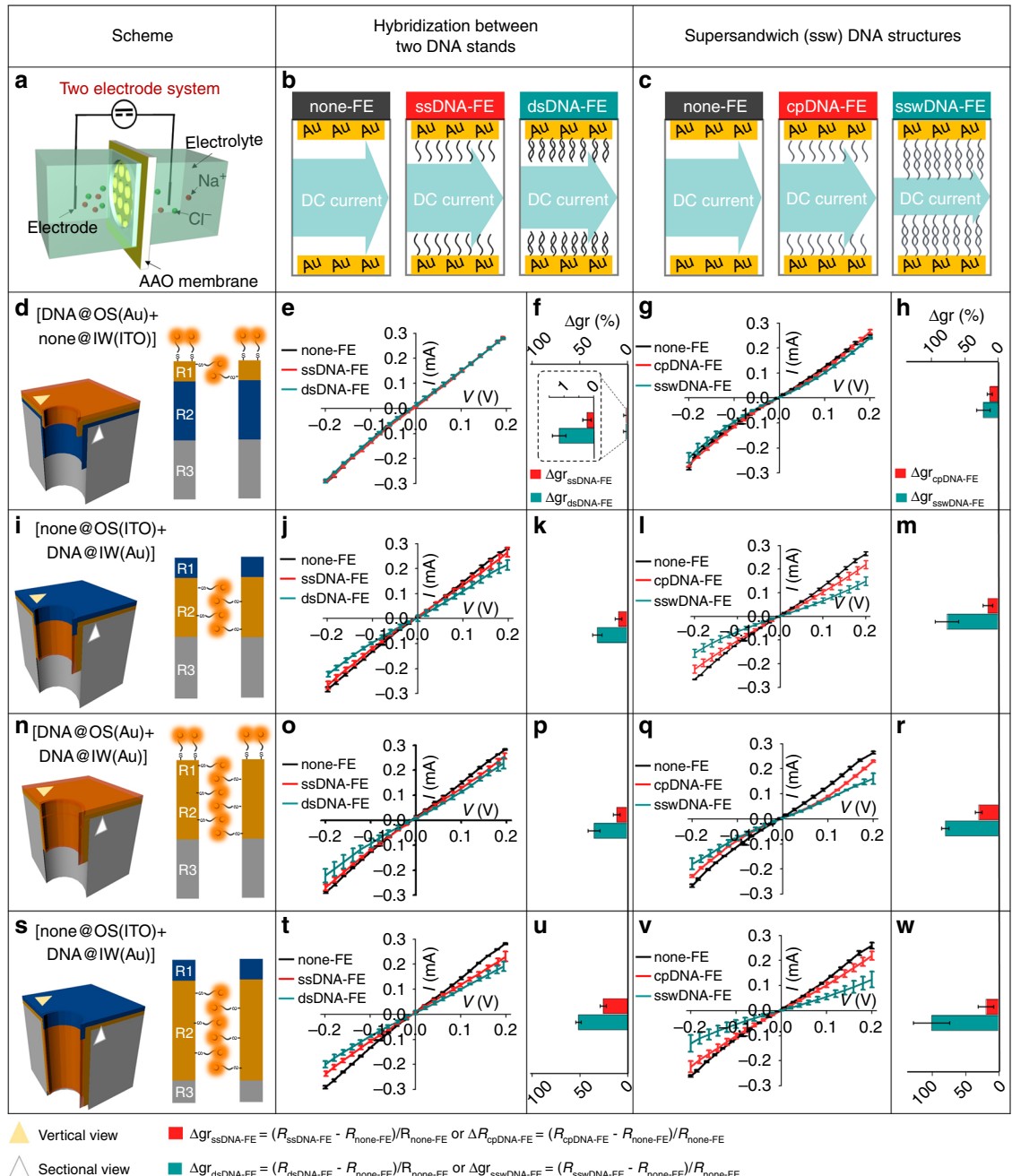

**Fig. 4** Measurements of ion gating. $\Delta gr_{DNA-FE}$, from four kinds of samples with varied $FE_{OS}$ and $FE_{IW}$. The I–V are applied to characterize the resistance of samples, reflecting the gating contribution of the IW and OS in a solid-state membrane after being modified with $FE_{IW}$ and $FE_{OS}$ on the IW and OS, respectively. **a** The cartoon of two-electrode system is used in electrochemical characterizations. Ions go through the nanochannels under the direct current (DC). There are several states of $FE_{OS}$ and $FE_{IW}$ at Au surface: **b** none-FE, ssDNA-FE grafting and dsDNA-FE grafting. **c** none-FE, cpDNA-FE grafting and sswDNA-FE grafting. Schemes illustrate the distribution of $FE_{OS}$ and $FE_{IW}$ at OS and IW, in four samples which are described in (**d**, **i**, **n**, **s**, respectively. I–V curves characterize the current of the state as: none-FE grafting, ssDNA-FE grafting and dsDNA-FE grafting in four samples which are described in **e**, **j**, **o**, **t**; none-FE grafting, cpDNA-FE grafting and sswDNA-FE grafting in four samples which are described in **g**, **l**, **q**, **v**. Varied $\Delta gr$ are defined in the bottom part. Calculated $\Delta gr$ are shown in **f**, **k**, **p**, **u** for DNA hybridization and **h**, **m**, **r**, **w** for sswDNA formation. Error bars represent standard deviations of the measured samples. Five experimental replicates for each data

electrodes which were placed in the two symmetric chambers filling with the tris buffer (pH = 7.4, 500 mM NaCl, 10 mM $MgCl_2$) as shown in Fig. 4a. The resistances under the three states were measured, as the following: 1) without FE ($R_{none-FE}$), 2) with FE ($R_{ssDNA-FE}$) and 3) with FE and its complementary DNA strand ($R_{dsDNA-FE}$). The following are the formulas:

$$\Delta gr_{ssDNA-FE} = (R_{ssDNA-FE} - R_{none-FE})/R_{none-FE}, \quad (1)$$

$$\Delta gr_{dsDNA-FE} = (R_{dsDNA-FE} - R_{none-FE})/R_{none-FE}. \quad (2)$$

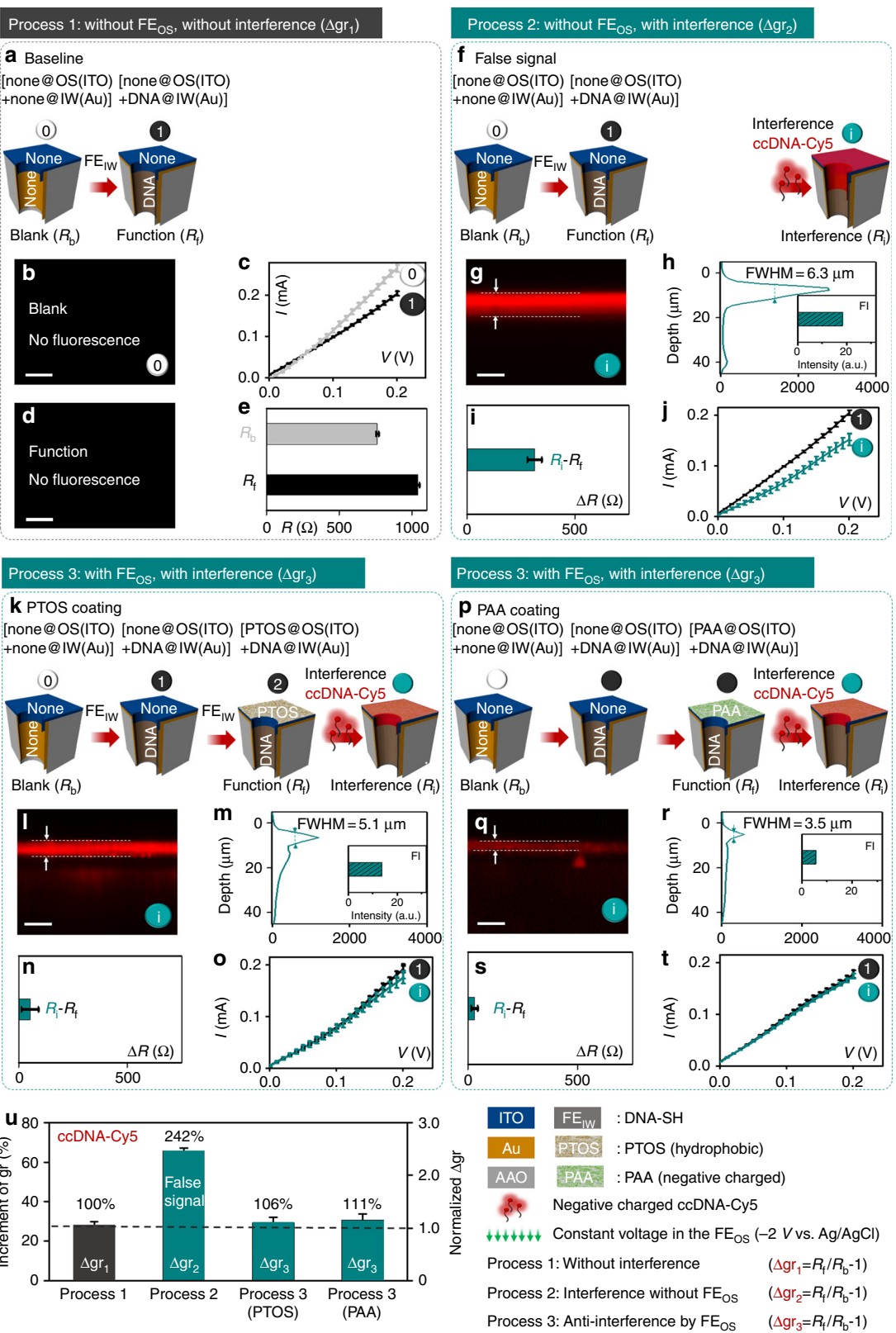

I–V responses were performed by using the sample with the pore size of $25 \pm 5$ nm and the thickness of 40 µm for dsDNA hybridization (Fig. 4e–t) and for sswDNA formation (Fig. 4g–v). The $\Delta gr_{ssDNA-FE}$ and $\Delta gr_{dsDNA-FE}$ in Fig. 4k–m are of $9.1 \pm 2.2\%$ and $31.6 \pm 2.8\%$, significantly greater than that in Fig. 4f–h ($0.24 \pm 0.13\%$ and $1.2 \pm 0.26\%$, respectively). Interestingly, the $\Delta gr_{ssDNA-FE}$ and $\Delta gr_{dsDNA-FE}$ in Fig. 4p–r are approximately equal to that in Fig. 4k–m, revealing the negligible contribution to the ionic gating from $FE_{OS}$ in Fig. 4p–r. Furthermore, the negligible contribution to the ionic gates from $FE_{OS}$ is further proved by using the deposition membrane with the larger pore size (Supplementary Fig. 8 and Supplementary Fig. 9).

**Fig. 5** The chosen $FE_{OS}$ endowing OS with the function, anti-interference. The $FE_{OS}$ is hydrophobic or negative molecular. Three different processes (Process 1 (**a**), 2 (**f**) and 3 (**k/p**)) were conducted to evaluate whether the $FE_{OS}$, we chosen, bring OS the anti-interference ability. cpDNA-Cy5 is chosen as the interference molecule. [none@OS(ITO) + none@IW(Au)] in Fig. 5a are in the 0 state in Process 1. [none@OS(ITO) + DNA@IW(Au)] in Fig. 5f are in the 1 state in Process 2. [PTOS@OS(ITO) + DNA@IW(Au)] in Fig. 5k are in the 2 state in Process 3. [PAA@OS(ITO) + DNA@IW(Au)] in Fig. 5u are in the 2 state in Process 3, respectively. Fluorescent images (**b**, **d**, **g**, **l**, **q**), fluorescent profile (**h**, **m**, **r**), resistances increment (**e**, **i**, **n**, **s**), I–V curves (**c**, **j**, **o**, **t**) are corresponding to the steps in Process 1, 2 and 3. **u** Comparison of the Δgr from the three processes mentioned above, and the following is the definition for them: $\Delta gr_1 = R_f/R_b - 1$ (for Process 1); $\Delta gr_2 = R_i/R_b - 1$ (for Process 2); $\Delta gr_3 = R_i/R_b - 1$ (for Process 3). Scale bars, 10 μm. Error bars represent standard deviations of the measured samples. Five experimental replicates for each data

Based on the above foundation, we endeavor to construct a high-efficiency ion gating system through generating DNA superstructure, wherein concatenated DNA strands repeatedly hybridized (Supplementary Fig. 10). These DNA superstructures are designed as FEs to amplify Δgr by enhancing steric hindrance in the nanochannels (with the pore size of 25 ± 5 nm and the thickness of 40 μm)[50]. The $\Delta gr_{cpDNA-FE}$ and $\Delta gr_{sswDNA-FE}$ in Fig. 4e, g are the smallest among the four patterns of the membranes (Supplementary Fig. 11). The $\Delta gr_{cpDNA-FE}$ and $\Delta gr_{sswDNA-FE}$ in Fig. 4p–r are approximately equal to the ones in Fig. 4k–m. The approximately equal $\Delta gr_{cpDNA-FE}$ or $\Delta gr_{sswDNA-FE}$ also suggests the negligible contribution of $FE_{OS}$ to ionic gating, similar to the conclusions in the last paragraph. When prolong the deposition duration of Au, the $\Delta gr_{cpDNA-FE}$ or $\Delta gr_{sswDNA-FE}$ is further enhanced (Fig. 4u–w). Furthermore, the negligible contribution of ionic gates from $FE_{OS}$ is further proved by using the deposited membranes with a high gating ratio up to 4635% and further applied in ATP detections (Supplementary Fig. 12 and Supplementary Fig. 13). Electrochemical impedance spectroscopy (in Supplementary Fig. 14 and Supplementary Fig. 15) also demonstrate that when linear DNAs act as the FEs for ionic gating[49], the impact from the $FE_{OS}$ are proved to be negligible, while the major contribution to ionic gating is resulted from the $FE_{IW}$ in our system.

The above conclusion seems to be inconsistent with our previous results, but it is not. We employed further calculations for the density of $FE_{OS}$ and $FE_{IW}$ at OS and IW (Supplementary Fig. 16), respectively, from our previous work. The amount of $FE_{IW}$ is about 7.2 times of the one of $FE_{OS}$, while the amount of $FE_{IW}$ is only 2.3 times of the one of $FE_{OS}$ in our previous work[45], which suggest that the ratio between the $FE_{OS}$ and $FE_{IW}$ is vital for the contribution to ion gating. The contribution from $FE_{OS}$ to ion gating in this work is negligible due to the much less ration between $FE_{OS}$ and $FE_{IW}$ than that in the previous work.

**The chosen $FE_{OS}$ endow OS with the function as anti-interference.** Since the $FE_{OS}$ in Fig. 4 are demonstrated to be negligible for the ionic gating, we propose to introduce $FE_{OS}$ to OS, which is expected to endow OS with a function (Fig. 5). In order to achieve the function of OS, the anti-interference is taken as an example, and then several different $FE_{OS}$ are chosen (Figs. 5, 6) to endow OS with the anti-interference function. Thus, 3 different processes were conducted to evaluate whether the $FE_{OS}$, we chose, bring OS the anti-interference ability (the cpDNA-Cy5 is chosen as the interference molecules in Fig. 5). There are three different processes: Process 1 (Fig. 5a, step 0 to step 1); Process 2 (Fig. 5f, step 0 to step i); Process 3 (Fig. 5k or p, step 0 to step i). The gating ratio ($\Delta gr_1 = R_f/R_b - 1$) in Process 1 (Fig. 5c–e) is the reference value, and the gating ratio ($\Delta gr_2 = R_i/R_b - 1$) in Process 2 (Fig. 5i, j) and ($\Delta gr_3 = R_i/R_b - 1$) in Process 3 (Fig. 5n–t) are compared with $\Delta gr_1$ to test the sample's anti-interference ability. The deviations from the test system can be ignored (Supplementary Fig. 17 and Supplementary Fig. 18). As expected (Fig. 5u), $\Delta gr_2$ is about 242% of the reference value ($\Delta gr_1$), which suggests that the sample in Fig. 5f without anti-interference ability, thus, induces the

false positive signal. In contrast (Fig. 5u), $\Delta gr_3$ for the samples with the $FE_{OS}$ (both perfluorooctyltriethoxy silane (POTS) in Fig. 5k and polyacrylic acid (PAA) in Fig. 5p) in Process 3 are around 106 and 111%, respectively, of the reference value ($\Delta gr_1$), which suggests that the samples with the $FE_{OS}$ endow OS with the function, anti-interference.

To explore the mechanism of the anti-interference for the OS with the $FE_{OS}$, the interference molecules are labeled with fluorescence molecules, Cy5. Almost no fluorescence in Process 1 (Fig. 5b–d). Both the intensity and the depth at step i in Process 3 (Fig. 5l–r) are weaker and thinner than the ones at step i in Process 2 (Fig. 5g, h), which means that the sample $FE_{OS}$ (PTOS and PAA) can block the interference molecules into the nanochannels, but the sample without $FE_{OS}$ cannot. It is possibly that the $FE_{OS}$, hydrophobic PTOS (Supplementary Fig. 19) and negative PAA (Supplementary Fig. 20), impede the hydrophilic and negative interference molecules (cpDNA-Cy5 chosen in Fig. 5) entering into the nanochannels. To further demonstrate the blocking mechanism, we performed the further destruction experiments, step d (in Process 2 and Process 3) after step i. The interference molecules were again added, and, this time, they were driven by −2 V constant voltage, entering into nanochannels. $\Delta R = R_d - R_i$ of the sample without $FE_{OS}$ (Supplementary Fig. 21e) at step d in Process 2 is set as the reference value, due to the sample without anti-interference ability. $\Delta R = R_d - R_i$ of the sample with $FE_{OS}$ (Supplementary Fig. 21e and Supplementary Fig. 21g) at step d in Process 3 are 98 and 85% of the above reference value, respectively, which further prove hydrophobic and negative $FE_{OS}$ blocking mechanism.

To further evaluate the conclusions in Fig. 5, we changed the interference molecules to AIEgens (Fig. 6), positive-charged small molecules, which are distinct from ccCy5-DNA in Fig. 5. Similar to the experiments in Fig. 5, here are three different processes: Process 1 (Fig. 6a, step 0 to step 1); Process 2 (Fig. 6f, step 0 to step i); Process 3 (Fig. 6k, step 0 to step i). The gating ratio ($\Delta gr_1 = R_f/R_b - 1$) in Process 1 (Fig. 6c–e) is the reference value, and the gating ratio ($\Delta gr_2 = R_i/R_b - 1$) in Process 2 (Fig. 6i, j) and ($\Delta gr_3 = R_i/R_b - 1$) in Process 3 (Fig. 6n, o) are compared with $\Delta gr_1$ to evaluate the samples' anti-interference ability. As expected (Fig. 6p), $\Delta gr_2$ is about 202% of the reference value ($\Delta gr_1$), which suggests that the sample in Fig. 6f without anti-interference ability. In contrast (Fig. 6p), $\Delta gr_3$ for the samples with the $FE_{OS}$ (Polyetherimide (PEI) in Fig. 6k) in Process 3 is around 143%, of the reference value ($\Delta gr_1$), which suggests that the sample endow OS with the function, anti-interference. The fluorescence results demonstrate that the sample $FE_{OS}$ (PEI) can block the interference molecules into the nanochannels, but the sample without $FE_{OS}$ cannot (Fig. 6b–m). It seems that the anti-interference performance for PTOS and PAA are better than that of PEI, which is probably due to the much larger size of interference molecules in Fig. 5 (DNA) than that in Fig. 6 (AIEgens). We will further do the comparisons in the Discussion part.

To explore the mechanism of the anti-interference for the OS with positive charged PEI as the $FE_{OS}$, we performed the similar

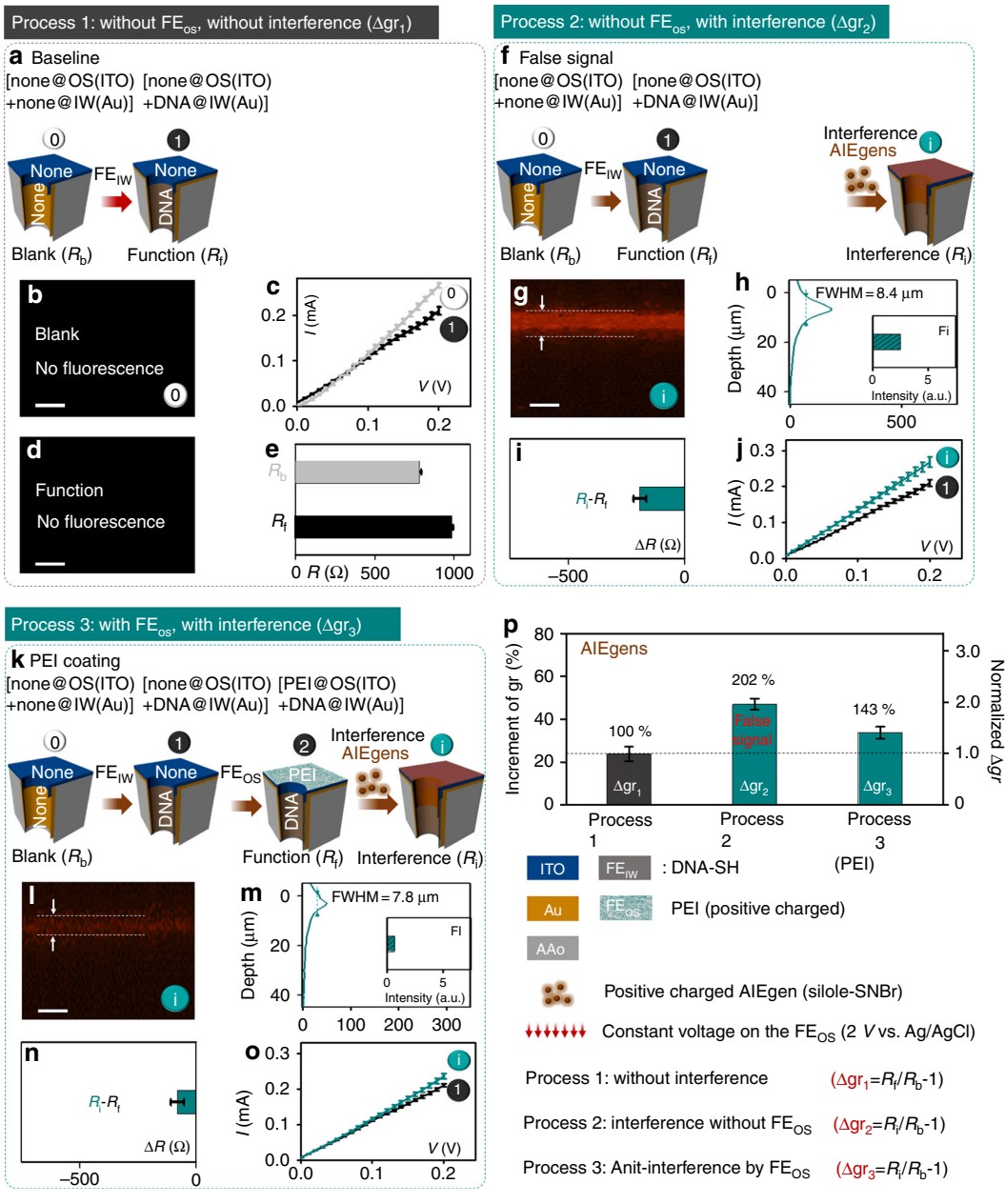

**Fig. 6** The chosen $FE_{OS}$ endowing OS with the function, anti-interference. The $FE_{OS}$ is positive molecular. Three different processes (Process 1 (**a**), 2 (**f**) and 3 (**k**)) were conducted to evaluate whether the $FE_{OS}$, we chosen, bring OS the anti-interference ability. AIEgen is chosen as the interference molecule. [none@OS(ITO) + none@IW (Au)] in **a** are in the 0 state in **Process 1**. [none@OS(ITO) + DNA@IW (Au)] in **f** are in the 1 state in Process 2. [PEI@OS (ITO) + DNA@IW (Au)] in Fig. 6k are in the 2 state in Process 3. Fluorescent images (**b**, **d**, **g**, **l**), fluorescent profile (**h**, **m**), resistances increment (**e**, **i**, **n**), I–V curves (**c**, **j**, **o**) are corresponding to the steps in Process 1, 2 and 3, respectively. **p** Comparison of the Δgr from the three processes mentioned above, and the definition for them: $\Delta gr_1 = R_f/R_b - 1$ (for Process 1); $\Delta gr_2 = R_i/R_b - 1$(for Process 2); $\Delta gr_3 = R_i/R_b - 1$(for Process 3). Scale bars, 10 μm. Error bars represent standard deviations of the measured samples. Five experimental replicates for each data

experiments with the ones in Fig. 5. It is possibly, that the $FE_{OS}$, positive PAA, impede the positive interference molecules (AIEgens chosen in Fig. 6) entering into the nanochannels. To further demonstrate the blocking mechanism, we performed the further destruction experiments, step d (Process 2 and Process 3 in Supplementary Fig. 22) after step i. The results of destruction experiments further prove positive $FE_{OS}$ blocking mechanism. Therefore, all of the chosen $FE_{OS}$, indeed, endow OS with the function, anti-interference to interference molecules (not only macro-molecules, but also small molecules; not only negative, but also positive molecules).

## Discussion

Since there are three different molecules, PTOS, PAA, and PEI, worked as $FE_{OS}$, respectively, how to select $FE_{OS}$ and how to evaluate their performance for anti-interference are two key questions. For the first one, the rules for selecting $FE_{OS}$ are: (1) $FE_{OS}$ would not react with the interference agents; (2) $FE_{OS}$ have the opposite chemical or physical properties with the interference agents, for example, (hydrophobic to hydrophilic; negative to negative and positive to positive). For the second one, we need to check the data in Figs. 5, 6 and Supplementary Fig. 21, 22 again, and the value of $R_{Destruction}$ for sample with PTOS as $FE_{OS}$ is

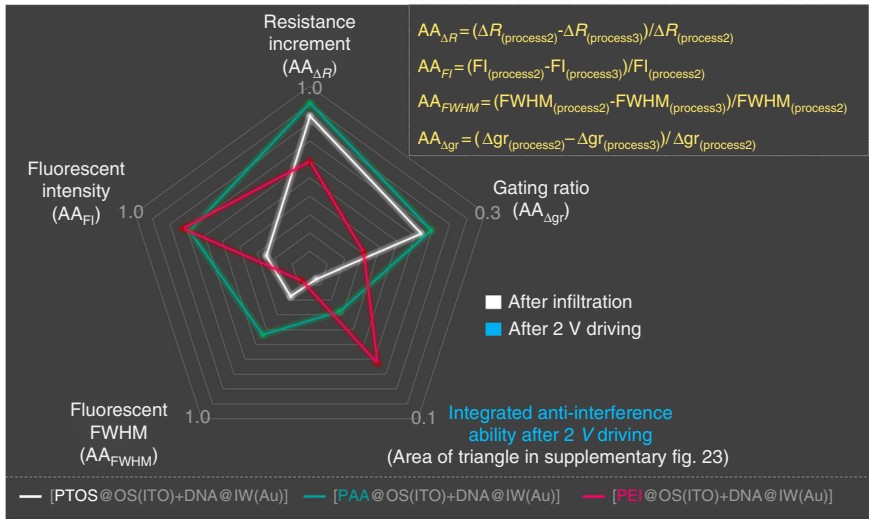

**Fig. 7** The radar map reflecting the anti-interference performance of $FE_{OS}$. The radar map integrates the five parameters reflecting the variation by the interference of cpDNA-Cy5 or AIEgens after infiltration and after 2 V driving. The five parameters are considered as the resistance increment from step 1 (or step 2) to step i ($AA_{\Delta R}$), gating ratio ($AA_{gr}$), fluorescent intensity ($AA_{FI}$), fluorescent FWHM ($AA_{FWHM}$). The calculated formulas are listed as $AA_{\Delta R} = (\Delta R_{(process\ 2)} - \Delta R_{(process\ 3)})/\Delta R_{(process\ 2)}$, $AA_{FI} = (FI_{(process\ 2)} - FI_{(process\ 3)})/FI_{(process\ 2)}$, $AA_{FWHM} = (FWHM_{(process\ 2)} - FWHM_{(process\ 3)})/FWHM_{(process\ 2)}$, $AA_{\Delta gr} = (\Delta gr_{(process\ 2)} - \Delta gr_{(process\ 3)})/\Delta gr_{(process\ 2)}$ (where the $\Delta gr$ were calculated by taking $R_B$ as 100%). The fifth parameter is the area of triangle in Supplementary Fig. 23 to value the integrated anti-interference ability after 2 V driving. The bigger area of the polygon, the greater anti-interference of OS

greater than the one for sample with PAA, which means blocking effect from PAA is superior to the one from PTOS. Both the intensity and the depth of sample at step 4 in Process 3 (Fig. 5q, r) are weaker and thinner than the ones of sample at step 4 in Process 3 (Fig. 5l, m), which also means that the blocking effect from PAA is superior to the one from PTOS. However, it is difficult to evaluate the effect among PAA, PTOS, and PEI due to different interference agents. We, therefore, designed a Radar map containing five parameters: the resistance variation from step 2 to step i, gating rate, fluorescent intensity, fluorescent FWHM and anti-interference ability after 2 V driving (calculated by area of triangle in Supplementary Fig. 23). The bigger area, the polygon is, and the greater anti-interference the OS perform (Fig. 7). Apparently, blocking effect from PAA is the best, and PTOS is superior to the one from PEI. It is possibly due to the fact that it is hard to block small molecules (AIEgens) compared with the macromolecules (cpDNA-Cy5).

Coincidentally, in a nuclear pore complex, the similar divisions of function exist naturally. Analogous to the OS, the densely packed Phe-Gly Nup meshwork at the entrance of nuclear pore complex physically exclude spurious macromolecules into nuclear pore complex (analogous to $FE_{OS}$), avoiding the interference on the gating properties of nuclear pore complex (Supplementary Fig. 24)[51]. Overall, this work could be to blaze a trial not only for the role of $FE_{OS}$ at OS in nanochannels, but even for the biomimetic system in the porous membrane.

## Methods

**Preparation of metallic deposited AAO membrane**. Firstly, the AAO membranes were immersed in the 1 M HCl solution under ultrasonic treatment for 2 min. Then, as-prepared AAO membranes were washed with distilled water and drying with nitrogen gas. Deposition has been performed by using AE Nexdap PVD platform (Angstrom Engineering Inc.)[45]. Two kinds of deposited target as Au and ITO were used respectively. The circular targets were approximately parallel to the AAO membranes, ensuring the deposition direction perpendicular to the membranes. The layer by layer depositions were taken by the successive deposition without replacing the target or releasing vacuum. The successive deposition ensured no secondary pollution at the first deposited layer. The extreme low deposited speed was applied as 0.01 nm s$^{-1}$. The deposition duration was 100 s, 500 s, and 1000 s, respectively. The deposited speed was calibrated by the deposited

thickness on the surface of the flat silicon wafer at nanometer level. The platform will calibrate the parameters automatically.

**DNA modification and hybridization**. Modification of thiol-modified DNA. The graft of thiol-modified DNA strings was achieved by immersing the Au deposited AAO membranes into a 5′-thiol modified DNA string (1 μM) of 10 nM tris solution (pH = 7.4, 500 mM NaCl, 1 mM MgCl$_2$) for 10 h[50]. The membranes were rinsed with distilled water and dried with nitrogen gas. The as-prepared membranes were then applied for further EC or LSCM characterizations. After the tests, the membranes were rinsed with distilled water and dried with nitrogen gas. For hybridization, the membranes grafting with single sequence string were then immersing into the DNA target (1 μM) for 10 h. The rinse and dry were done.

Modification of amino-modified DNA. The membranes were washed with distilled water and dried in argon gas. After that, the membranes were immersed into a 5% acetone solution of APTES for 10 h. The membranes were thoroughly washed in acetone and baked at 120 °C for 2 h, and left overnight in 25% aqueous solution of glutaraldehyde (25 wt% aqueous solution). After thoroughly washing with distilled water and drying with nitrogen gas, the membranes were modified with 5′-aminated DNA (the capture probe, 1 μM) in 1 ml of 10 mM Tris solution (pH = 7.4, 500 mM NaCl, 1 mM MgCl$_2$) for 10 h[50]. Furthermore, the modification of two DNA probes was achieved by immersing the salinized membranes into the equimolar amino-modified and thiol-modified DNA (1 μM) for 10 h. The membranes were rinsed with distilled water and dried with nitrogen gas.

DNA hybridization and the formation of DNA supersandwich structure. For hybridization, the membrane grafting with single sequence string was immersed into the DNA target (1 μM) for 10 h. The DNA supersandwich structure was formed by immersing the cp-DNA modified DNA into the mixture of equimolar P1 and P2 (1 μM) for 10 h. All the membranes after DNA hybridization were rinsed with distilled water and dried with nitrogen gas after the DNA hybridization.

**Superficial functionalization ($FE_{OS}$@OS)**. The perfluorooctyltriethoxysilane (POTS) coating was achieved by adding the PTOS solution on the side by metallic deposition for 30 min and rinsing throughout by alcohol. The PAA and PEI coating were achieved by the spin-coating of PAA solution (1 mM) and PEI solution (1 mM) at 3000 rpm. The spin-coating carried out for twice, each time adding 200 μl and lasting for 15 s. After the spin-coating, the membranes were washed throughout by water.

**Time of flight secondary ion mass spectrometry (TOF-SIMS)**. Secondary ion mass spectra of deposited AAO surfaces were characterized by TOF-SIMS V (IONTOF, GmbH). A Bi liquid metal primary ion source was applied with an angle of 45° relative to the sample surface with a pulsed Bi$^{3++}$ primary ion beam of 30 keV and shave off fresh 60 μm × 60 μm areas for each analysis. The TOF analyzer was installed at an angle of 90° to the sample surface. Negative secondary ion spectra were collected. Mass calibration was carried out using standard procedures (mass resolving power > 5000).

**Laser scanning confocal microscope (LSCM).** The fluorescent-dyed AAO membrane were clamped by cover glass and microscope slide filling with water (around 1 cm² AAO with 100 μl water). The LSCM characterizations were performed by using LSCM system (Olympus FV1200) with a 40× objective. For FITC-modified DNA, excitation: 495 nm and emission collected: 505–535 nm. For Cy5-modified DNA, excitation: 565 nm and emission collected: 655–685 nm. For Silole-SNBr (AIE), excitation: 466 nm and emission collected: 565–595 nm. To characterize the fluorescent distribution along Z-axil, Z-axil scanning was applied by 500 nm for each step. On purpose of comparison of fluorescent intensity, the settings of channel parameter were constant, including HV = 600, Gain = 1.75 and Offset = 1. All membranes were avoided light before LSCM tests.

**Electrochemical measurements.** The electrochemical characterizations of the AAO and its derivative products were performed by using two-electrode system in a 10 mM tris solution (pH = 7.4, 500 mM NaCl, 1 mM MgCl₂) as electrolytes. The symmetric Ag/AgCl electrodes were used as a working and a counter electrode. Bio-logic VMP3 station (Bio-logic Science Instrument Pvt Ltd) was used to measure the electrochemical responses. The I–V tests were performed in the potential from −0.2 to 0.2 V under a direct-current mode. The EIS measurements were performed under an alternating-current mode. The tests were performed in a frequency range from 1 MHz to 10 mHz at the open circuit potential with an AC perturbation of 5 mV. The electric driving was performed under constant voltage as −2 or +2 V for 10 min. Before EC measurement, each dry membrane was immersed in 10 nM tris buffer overnight to reduce the test error coming from inadequate impregnation.

## Data availability

The data that support the findings of this study are available from the corresponding author on reasonable request.

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

## Acknowledgements

This work is supported by the National Basic Research Program of China (973 Program, 2015CB932600), the National Key R&D Program of China (2017YFA0208000, 2016YFF0100800), the National Natural Science Foundation of China (21525523, 21722507, 21574048, 21605053, 21874121, 21802130), The Fok Ying-Tong Education Foundation, China (151011). This work is supported by the open project of Hubei Key Laboratory of Forensic Science (No. 2018KF001).

## Author contributions

F.X. designed and directed the project. P.G. fabricated devices, performed measurements and carried out data analysis with help from Q.M., D.D., D.W., T.Z., and X.L. P.G., D.D., and F.X. wrote the manuscript. All authors contributed to discussions.

## Additional information

**Competing interests:** The authors declare no competing interests.

