## [Peer Review File · Nature Communications]

Response to Reviewer 1# (Q, the referee's comments; A, the authors' response)

Major changes in the revised manuscript are highlighted with blue color.

Reviewer 1#: In the manuscript “Role of outer-surface-probe for blocking interference molecules and inner-wall-probe for regulating ion gating of nanochannels” Pengcheng Gao et al. report the effect of functional elements (FEs) on ionic gating and anti-interference. Design of nanochannels with cooperative outer surface (OS) and inner wall (IW) FEs is of great importance for broad applications. The results presented here are promising and substantial. I would recommend the publication of the manuscript after minor revisions. My minor comments are listed below.

Thanks a lot for the Reviewer 1# for his/her positive comments on our paper. He/ She comments our paper as “of great importance for broad applications” and “promising and substantial”. We also highly appreciate these constructive suggestions that are of great significance to improve our manuscript.

Q1: As the authors have shown that the FE_{OS} plays a negligible role on ionic gating but rather blocks the interfering molecules, the term “outer-surface-probe” in the title might be misleading.

A1: We appreciate the Reviewer 1#' s suggestion. Now we replace the term “outer-surface-probe” in the title with “function-element@outer-surface” and the term “inner-wall-probe” in the title with “function-element@inner-wall”. We also confirm that there is not the expression as “outer-surface-probe” or “inner-wall-probe” in the manuscript.

Q2: The analog between FE_{OS} of the synthetic system designed by the authors and the biological lipid bilayer (Scheme S1) is not very appropriate. Also, there is a misconception in the manuscript “lipid bilayers with a hydrophobic outer surface”, as the surface of the lipid bilayer that is exposed to water is hydrophilic. A better example of biological nanochannel with potential blocking biopolymers at its entrance could be the nuclear pore complex.

A2: We agree with the Reviewer 1# 's comment “The analog between FE_{OS} ...and the biological lipid bilayer is not very appropriate”. As the Reviewer 1# 's suggestion, we carefully learnt many related publications about the “nuclear pore complex”, and were very glad to find that the “nuclear pore complex” recommended by the Reviewer 1# is indeed similar to the nanochannel system in the present work. Firstly, nuclear pore

complex itself possesses densely packed Phe-Gly Nup meshwork at the entrance of the nuclear pore complex. “The FG(Phe-Gly) Nups may exhibit a polymer brush-like behaviour, pushing away nonspecific macromolecules.” (*Nat. Rev. Mol. Cell. Bio.* **11**, 491-501(2010)), which is analogous to the appearance form and function of FE_{OS} in our nanochannel system. Secondly, the Phe-Gly Nups at the inner wall of nuclear pore complex regulate the transport of substrates, analogous to that of FE_{IW} in the present work. Accordingly, now we redraw the Scheme S1 based on the analogue above, simultaneously refer to the descriptions in *Nat. Rev. Mol. Cell. Bio.* **11**, 491-501(2010), as shown in **Scheme R1**. The corresponding corrections are done in the manuscript and the supplementary information, highlighted in blue.

Scheme R1. The analogous divisions of function exist naturally in a nuclear pore complex. In a nuclear pore complex, the Phe-Gly Nup meshwork at the entrance of nuclear pore complex (analogous to FE_{OS}) physically excludes macromolecule from the nuclear pore complex. Meanwhile, the Phe-Gly Nup inside nuclear pore complex (analogous to FE_{IW}) regulates the transport of substrates through nuclear pore complex. Produced with permission from *Nat. Rev. Mol. Cell. Bio.* **11**, 491-501(2010), Copyright [2010], Springer Nature. All right reserved.

Q3: The language of the manuscript could be improved. Some sentences are of grammar issues. “Scientists extracted or biommitic synthesized...” “It is possibly due to it is hard to block ...”

A3: Thanks a lot for the Reviewer 1# 's careful reading and the grammar issues have been corrected as Reviewer 1# 's suggestions. Not only this, we do more corrections to improve the language throughout the manuscript. Grammar mistakes have been listed in the response letter. The articles, tense, singular & plural have been checked throughout the manuscript. All corrections are highlighted in blue.

Page 1, Line 15 “researchers focus all interests on...” instead of “virtually all interests of nanochannels focus on...”

Page 2, Line 33 “Scientists modify the nanochannels, obtained by extraction or biomimetic synthesis, with...” instead of “Scientists extracted or biomimetic synthesized...”

Page 2, Line 40 “theory simulations have forecasted...” instead of “theory simulations predicted...”

Page 2, Line 44 “If we failure to settle this issue, it will be an obstacle to the...” instead of “A failure to settle the issue has been an obstacle to fulfill...”

Page 3, Line 58 “nonspecifically” instead of “non-specifically”

Page 3, Line 12 “sole FE_{IW} ” instead of “solely of FE_{IW} ”

Page 7, Line 142 “The measurement depth of Au from fluorecence is larger than...” instead of “The data for the depth of Au from fluorecence is more than...”

Page 10, Line 185 “after being modified with” instead of “after modifying FE_{IW} and FE_{OS} ”

Page 13, Line 256 “ FE_{OS} cannot block” instead of “ FE_{OS} could not”

Page 15, Line 283 “further do” instead of “continue”

Page 18, Line 340 “It is possibly due to the fact that it is hard to block ...” instead of “It is possibly due to it is hard to block ...”

Response to Reviewer 2# (Q, the referee's comments; A, the authors' response)

Q1: The Authors start the manuscript with motivation for their work that is focused on DNA sequencing. I do appreciate the importance of understanding the role of the outer surface of the membrane for DNA sequencing and other molecules detection, but the system the Authors study is very different from nanopore systems used for molecular sensing. For example, the system they study has much larger diameter and thickness than any other nanopore system used for DNA analysis.

A1: We are very grateful to the Reviewer 2# 's positive comments as "importance of understanding the role of the outer surface of the membrane". And we also thank the Reviewer 2# 's comments about the different nanopore systems. However, we would like to politely point out that our motivation in this paper is absolutely not the DNA sequencing, but exploring the new function of the FE_{OS} on OS of nanochannels and differentiating the functions of the FE_{OS} and the FE_{IW} on IW of nanochannels. Namely, FE_{IW} are serving for ionic gating and the chosen FE_{OS} decrease the false signals for the ionic gating in complex environments by blocking interference molecules into the nanochannels. The DNA hybridization applied in our work is to construct an ionic gating case. And the DNA sequence is not the key issue in this work.

As for our nanochannels are different from nanopore systems used for molecular sensing, we believe that both nanochannel based sensing and nanopore based sensing are the recently advanced sensing method. And the two sensing methods have in common with that both of the two use a two-electrode system to detect the variation of transmembrane ionic current brought by adding analytes. (*Chem. Soc. Rev.* **38**, 2360-2384(2009), *Angew. Chem. Int. Ed.* **52**, 13154-13161(2013), *Chem. Soc. Rev.* **47**, 322-356(2018)) (All scheme of working principles are listed in **Figure R1**). Many pervious works have been done by using nanochannels for molecular sensing in addition to nanopores (*Nano Lett.* **7**, 1609-1612 (2007), *Anal. Chem.* **89**,1110-1116 (2017), *Nanotechnology*, **19**, 485711(2008), *ACS Nano* **10**, 7476-7484 (2016)). And we also published several papers by using nanochannels for molecule sensing including ATP and DNA detections (*Angew. Chem. Int. Ed.* **52**, 2007-2011 (2013)), glucose detections (*NPG Asia Mater.* **8**, e234 (2016)), Zn^{+} ion detections (*Analyst* **141**, 3626-3629 (2016)), DNA detections (realized the single-base mismatch detections of DNA) (*Adv. Mater.* **28**, 460-465 (2016)). Although the nanochannels are larger in diameter and thickness than the nanopores, it generally does not impact on the application of nanochannels in molecular sensing.

Inspired by Reviewer 2# 's comments that our nanochannel system is also promising in molecular sensing, we add a group of experiments about ATP detection by our system. The ATP will trigger the disassembly of supersandwich DNA structure (ssw-DNA) and further leads to the rise of transmembrane ionic current. The variation of ionic currents can be used for ATP detections. The detection of ATP has been achieved by using our system with sensitivity to 10 nM. Meanwhile, our system shows higher selectivity towards its target (ATP) than the other three types of nucleoside triphosphates as CTP, GTP, and UTP, as shown in Figure R2. The experiment details are added in the Supplementary Methods 1.2, and Figure R2 is added in the Supplementary Information as Supplementary Fig. 13. The AAO membranes, from the same company with the similar pore diameter and the same thickness, were also used for molecular sensing as ATP detections in our previous work. The parameter of the AAO membranes used in this experiment is listed in Table R1.

Reference	Scheme of working principle	Reference	Scheme of working principle
Chem. Soc. Rev. 38 , 2360-2384 (2009). In Figure 1		Chem. Soc. Rev. 47 , 322-356 (2018). In Figure 29	Angew. Chem. Int. Ed. 52 , 13154-13161 (2013). In Figure 1		Present work In Figure 3	
Figure R1. the scheme of working principle in the present work and the work of others.

Table R1

	Materials	Diameter	Thickness	Company
J. Am. Chem. Soc. 134 , 15395-15401 (2012)	Anodic aluminum oxide	56 ± 3 nm	60 μm	PuYuan Nano
Nat. Commun. 9 , 40 (2018).	Anodic aluminum oxide	40-70 nm	60 μm	PuYuan Nano
Present work	Anodic aluminum oxide	25 nm	40 μm	PuYuan Nano

Figure R2. Detection of ATP using DNA supersandwich-modified nanochannels in the present system. (A) I-V curves before and after ATP treatment with different concentration from 1 nM to 1 mM. CP: nanochannels modified with the capture probe of ssw-DNA structure, SSW: nanochannels modified with the ssw-DNA structure. (B) Dose-response curves. The signal increase is defined as $(I_{ATP} - I_{SSW}) / I_{SSW}$, where I mean the current at 2.0 V. (C) Our system is highly selective for ATP, in contrast with other three types of nucleoside triphosphate (NTP), CTP, GTP, and UTP. (D) The signal increment of the I-V curves in Fig. R2.

Q2: Two electrode system is used to probe ionic transport through highly porous, low resistance membranes. Due to large number of pores, the resistance of the membrane is comparable to the resistance of the bulk electrolyte thus the voltage applied drops not only across the membrane but also in the solution. Consequently, the voltage values shown in current-voltage curves (or for driving molecules) are not physically meaningful.

A2: We agree with the Reviewer 2# 's comments that the resistance of the membrane in our two-electrode system is comparable to the resistance of the bulk electrolyte, due to large number of pores. Thus the voltage applied drops across both membrane and solution. And thanks a lot for Reviewer 2# 's careful reading. However, I'm afraid we can't agree with Reviewer 2# that "the voltage values shown in current-voltage curves (or for driving molecules) are not physically meaningful". It should be first pointed out that the voltage variation is the key in these current-voltage curves rather than the absolute voltage values. What will give rise to the voltage variation? That is the key point!

In fact, we have already considered that the high electrolyte resistance might bring

some bad effects on the voltage applied drops just like Reviewer 2# 's concern. But we solved this problem by establishing a testing system with tiny variation of electrolyte resistance (R_s) based on EIS spectra. As shown in Figure S13, the variation of electrolyte resistance is little (changed within 5% in Table R2) between each test from the EIS results. Accordingly, the electrolyte did not play a role in the voltage variation in current-voltage curves in spite of its high resistance in the present work. Hence, it is membrane state (such as 1* membrane with FEs, 2* membrane with FEs and interferences and 3* membrane with FEs and Targets) that gives rise to the voltage variation. Since the voltage variation is almost completely resulted from membrane states, the voltage values shown in current-voltage curves in our system are definitely physically meaningful.

We appreciate the Reviewer 2# 's concern about the variation of electrolyte resistance during the process of driving molecules. Thereby, a new group experiments have been done to test the R_s during the process for driving molecules under 2 V driving (Figure R4), which is added in the Supplementary Information as Supplementary Fig. 18. According to the EIS results, the R_s under either 0 V or 2 V kept almost constant for the three kinds of membranes modified with different FE_{OS} (the membranes modified with PTOS as FE_{OS} , the membranes modified with PAA as FE_{OS} and the membranes modified with PEI as FE_{OS}) used in Figure 4 and Figure 5. Hence, we believe that the voltage values for driving molecules are also physically meaningful.

Table R2 Calculated the R_s and the R_s variation from the EIS data of Figure S13

	R_s (none-FE)	R_s (ssDNA-FE)	R_s (dsDNA-FE)	$[R_s$ (ssDNA-FE) - R_s (none-FE)] / R_s (none-FE) (%)	$[R_s$ (dsDNA-FE) - R_s (ssDNA-FE)] / R_s (ssDNA-FE) (%)
	(Ω)	(Ω)	(Ω)		
Fig. S13e	579	596	585	2.9	1.9
Fig. S13j	544	562	546	3.3	2.9
Fig.S13m	540	545	567	0.9	4.0
Fig. S13p	547	558	558	1.4	0.5
	R_s (none-FE)	R_s (cpDNA-FE)	R_s (sswDNA-FE)	$[R_s$ (cpDNA-FE) - R_s (none-FE)] / R_s (none-FE) (%)	$[R_s$ (sswDNA-FE) - R_s (cpDNA-FE)] / R_s (cpDNA-FE) (%)
	(Ω)	(Ω)	(Ω)		
Fig. S13g	618	632	662	2.2	4.7
Fig. S13k	598	599	605	0.2	1.0
Fig. S13h	609	611	618	0.3	1.1
Fig. S13q	603	605	614	0.3	1.4

Figure R3. (A) EIS sketch and equivalent circuit. Top figure is the EIS sketch labelling the impedance elements in different frequency, Bottom figure is the equivalent circuit used to simulate the impedance spectra from Supplementary Fig. 15. (B) The statistic of the solution resistances (R_s) from the EIS data in Figure S13.

Figure R4. The EIS spectra of the three membranes in Fig. 4 and Fig. 5 [Grey line: PTOS@OS(ITO)+DNA@IW(Au), Green line: PAA@OS(ITO)+DNA@IW(Au) and Red line: PEI@OS(ITO)+DNA@IW(Au)]. (A) at 0 V and (B) at 2 V, respectively. The enlarge figure is to show that the solution resistance (R_s) of three membranes is approximately equal.

Q3: The gating effects that are reported are small.

A3: We agree with the Reviewer 2# 's comments that our gating effects are not as large as others'. However, in the present work, our purpose is to explore the new function of the FE_{OS} of nanochannels and realize the function partition of the FE_{OS} and FE_{IW} of nanochannels rather than developing a system with a high gating event. And we have discussed that the gating effects of the nanochannels are dominated by FE_{IW}, while the FE_{OS} block interference molecules into the nanochannels and decrease the false signals for the ionic gating in complex environments. We also proved that the FE_{OS} play little role in the gating effects, please see main body section "Contribution from FE_{OS} and FE_{IW} to ionic gating".

Given the Reviewer 2# 's concern about the small gating effects, we design a new system to achieve much larger gating effects than those in the present work by redesigning the FE_{IW} of nanochannels (Experiment details are added in the Supplementary Methods 1.2). After forming supersandwich DNA structure on inner wall of nanochannels, the resistance of the new designed membrane increases from 6.8 k Ω to 322 k Ω in Figure R5 C. The increment of gating ratio (Δgr) is 4635% ($\Delta gr = R_{ssw}/R_{cp} - 1$), which is far greater than that in manuscript (the maximum Δgr in the manuscript is 101%). Moreover, to investigate the contribution of FE_{OS} to the ionic gating, three kinds of deposited membranes are synthesized within a comparable system including the membrane in Figure R5(C). The deposited membranes are further modified by supersandwich-DNA structures as FEs, which are divide into three categories as the membrane modified with only FE_{OS} in Figure R5(A), the membrane modified with both FE_{OS} and FE_{IW} in Figure R5(B) and the membrane modified with only FE_{IW} in Figure R5(C). The Δgr of the membrane with only FE_{OS} is of 4.0 % in Figure R5(A), significantly smaller than the ones with both FE_{IW} and FE_{OS} in Figure R5 (B) ($\Delta gr = 3950\%$) and the membrane with only FE_{IW} in Figure R5 (C) ($\Delta gr = 4635\%$). Simultaneously, even having additional FE_{OS}, the Δgr of the membrane with both FE_{IW} and FE_{OS} isn't larger than the Δgr of the membrane with only FE_{IW}. All above, the contribution of FE_{OS} to gating effects is still negligible, though the system with much larger gating effects. Fig. R5 is added in the Supplementary Information as Supplementary Fig. 12.

Actually, we have already concerned that if the FE_{OS} influence the gating effect when the gating effects were larger or smaller in the present work. Three systems with different gating effects, thus, have been investigated, which are the systems with smaller gating effects in Figure 3 E, J, M and P, with middle gating effects in Figure S9 and with larger gating effects in Figure 3 G, K, N, Q. All experiments certified that the influence from FE_{OS}

on various gating effects is negligible in the present work. Therefore, even though the large gating effects are of significance, it is not the key issue in this work. But it is worth exploring in our next step!

Figure R5. Left figures: The I-V curves of three kinds of membranes with different deposited sequences after modified with capture probe (red line) and after modified with supersandwich DNA structure (green line). CP: nanochannels modified with the capture probe of supersandwich DNA structure, SSW: nanochannels modified with the supersandwich DNA structure. Right figures: The corresponding resistance calculated from the I-V curves.

Q4: The figures are too small, too complicated, and unclear.

A4: We appreciate the Reviewer 2# 's suggestion and redraw the Figure 4 and Figure 5. The new Figure 4 and Figure 5 can express the main content of the previous figures. The pervious Figure 4 and Figure 5 have been removed to supplementary information as Figure S19 and Figure S20. The corresponding correction has been done in the main body and the supplementary Information highlighted in blue.

Q5: The excessive amount of abbreviations makes the manuscript extremely hard to read.

A4: We are genuinely thankful for the Reviewer 2# 's kind remind about the abbreviations. Since the work refers to a large number of experiments and calculations, a number of abbreviations were used. According to the Reviewer 2# 's suggestion, we delete several unfrequented abbreviations in the manuscript, including EDS, In, EIS, $AA_{\Delta R}$, $AA_{\Delta gr}$, AA_{Fl} , AA_{FWHM} . Simultaneously, we draw a new table as Table S3 with all abbreviations in supplementary information to help read the manuscript. All corrections are highlighted in blue.

Table S3 Abbreviations

No	Abbreviation	Full name
1	FE	Functional element
2	IW	Inner wall
3	OS	Outer surface
4	FE_{IW}	FEs at the inner wall
5	FE_{OS}	FEs at the outer surface
6	TOF-SIMS	Time of flight secondary ion mass spectrometry
7	FITC	Fluoresceine isthiocyanate
8	Cy5	Cyaine-5
9	LSCM	Laser scanning confocal microscopes
10	Δgr	Increment of gating ratio
11	ss-DNA	Single-strand DNA
12	ds-DNA	Double-strand DNA
13	cp-DNA	DNA capture probe of supersandwich DNA structure
14	ssw-DNA	Supersandwich DNA structure
15	cc-DNA	Complete complementary DNA
16	EIS	Electrochemical impedance spectroscopy
17	POTS	Perfluorooctyltriethoxy silane
18	PAA	Polyacrylic acid
19	PEI	Polyetherimide
20	AA	Anti-interference ability

Response to Reviewer 3# (Q, the referee's comments; A, the authors' response)

Q1: This paper reports the role of outer-surface (membrane surface) for nanopore sensors. In this paper, the outer surface probes were used for blocking interference molecules. But if the objective is just to block interference molecules, it could be a good option to use a different membrane filter.

A1: We are very grateful to the Reviewer 3# 's careful reading and suggestions. But we can hardly agree to that "it could be a good option to use a different membrane filter".

First, our motivation of designing FE_{OS} is not "just to block interference molecules", but to blaze a new trail for the role of FE_{OS} at OS in nanochannels. To block interference molecules is just an instance to demonstrate the independent function of FE_{OS} differed from the gating effects of the FE_{IW} . And the Reviewer 1# 's comments, "Design of nanochannels with cooperative outer surface (OS) and inner wall (IW) FEs is of great importance for broad applications" shed light on the research value.

Second, it is not easy to find such a fit membrane filter for blocking interference molecules in a specific case. Let's take the case in the present work for an example. The interferences in the present work is DNA (with 27 bps \approx 9 nm length and 2 nm diameter) or AIEgens (MW = 965.3). If we block such small interferences with a membrane, the membrane filter either should be with a far smaller pore size than that used in the present work or should rely on such as electrostatic or hydrophobic repulsion. The membrane with such smaller pore size, such as below 2 nm, is neither facile to be fabricated nor easy to be modified for further application such as ionic gating. Additionally, the electrostatic or hydrophobic repulsion from the nanochannels will severely disrupt the ionic gating in nanochannels especially that with a small diameter. On the contrary, in the present work, we block the interferences by FE_{OS} , and we are not demanding about the diameter of nanochannels of a membrane. Besides, the FE_{OS} scarcely impact on the gating effects of FE_{IW} in nanochannels. Moreover, FE_{OS} can block various interferences such as hydrophobic FE_{OS} to block hydrophilic interference, and electric charged FE_{OS} to block the charge interference including the positive-charged FE_{OS} to block positive-charged interferences and the negative-charged FE_{OS} to block negative-charged interferences. But it is difficult to fabricate one membrane that can block various interferences like the present work unless we are very lucky to find the right membrane with both small pore size and electrostatic/hydrophobic repulsion. Even though we are very fortunate to find the multi-function membrane filter for blocking various interference molecules, it is also a good option to further modify the membranes with FE_{OS} , which could block more types of interferences over the membrane filter itself could. It will advance or diverse the functions

of the membrane.

Third, there is also another option to introduce a new membrane filter instead of FE_{OS} for blocking interference molecules. One filter membrane blocks interference, the other acts as ionic gating. However, to block interference molecules, the FE_{OS} have two advantages over the combination of two filters:

(1) Avoiding leakage of analyte. The gap between the two filters will lead to the leakage of analyte, especially for small molecules, such as ions, biomolecules. To avoid leakage, a complex system must be established for the two filters. In contrast, in our system, the FE_{OS} is directly bonded to the outer surface of nanochannels. The analyte can pass through FE_{OS} and then directly enter into inner wall, which never results in leakage.

(2) Reducing residence of analyte in the region of blocking interference. The filters are usually with a thickness up to hundreds of nanometers or even micrometer in consideration of mechanical strength. Not only interferences but also targets will be stuck in such a long pathway of the filters. In contrast, in our system, the thickness of FE_{OS} can be well controlled in nanoscale, below 100 nanometers. Compared with multiple filters, the shorter path of FE_{OS} will avoid the residues of targets to a large extent.

To sum up, the motivation of the present work is to demonstrate the independent function of FE_{OS} differed from the gating effects of the FE_{IW} . To block interference molecules is an instance for the new function of FE_{OS} . Moreover, for blocking interference molecules, we would like to politely point out that the FE_{OS} has advantages over a different membrane filter as discussed above.

Q2: Furthermore, this paper is similar to the authors' recent published paper, "Role of outer surface probes for regulating ion gating of nanochannels," *Nat. Commun.* 9, 40 (2018). The objective might be different but the contents are similar. I hesitate to recommend this paper for publication in *Nat. Commun.*

A2: Thanks a lot for the Reviewer 3# 's attention about our recent published paper and we also appreciate the Reviewer 3# 's scientific and logical thinking to compare these two works. However, I'm afraid we can't agree with the comments of Reviewer 3# that "this paper is similar to the authors' recent published paper". This is not the case! We designed and did a large number of new experiments to achieve the absolutely different objective in the present work. They definitely make the contents of the two papers very different. The objective of our previous works is to investigate the contribution of FE_{OS} and FE_{IW} to a single function as ionic gating. In the present work, our objective is to explore the new function of the FE_{OS} , the FE_{IW} and FE_{OS} are with two independent functions.

In the present work, our objective is to endow the FE_{IW} and the FE_{OS} for independent functions in one nanochannel system. Hence, the explicit region partition in a nanochannel system are highly desired, where outer-surface is one region modified by FE_{OS} and inner wall is the other region modify by FE_{IW} , respectively. To achieve this objective, we employed a time of flight secondary ion mass spectrometry (TOF-SIMS) to characterize the distribution of Au and ITO along nanochannels for the first time, the Au serving as IW and the ITO serving as OS, rather than EDS used in the previous work. It should be point out that the accuracy of the elemental recognition of TOF-SIMS is much better than that of EDS mapping. Moreover, TOF-SIMS can offer a 3D distribution of elements, which can restore the real appearance of nanochannel system such as the position of Au coating and ITO coating and the thickness of them (as shown in Figure 2). In contrast, the EDS mapping can only offer a 2D distribution of elements, it can't meet our demand any more, though it is enough in the previous work. The information of TOF-SIMS is more accurate and comprehensive than that of the EDS mapping. That is crucial to precisely divide the regions in a nanochannel system.

Furthermore, we investigated the distribution of the different FEs in one nanochannel system. In contrast, the distribution of the FEs wasn't investigated in the previous work. In order to characterize the distribution of FEs, we synthesized thiol-modified DNAs with Cy5 fluorescent and amino-modified DNAs with FITC fluorescent. The thiol-modified DNAs specifically bonding on Au through Au-thiol binding, while amino-modified DNAs specifically bonding on ITO through silane coupling reactions. The two DNA fluorescents have different emission, which can be characterized by Laser scanning confocal microscopes (LSCM). Accordingly, as far as we know, for the first time, we successfully confirmed that the thiol-modified DNAs and amino-modified DNAs are indeed bonding on the OS and IW of nanochannel system, respectively (as shown in Figure 3). It is the foundation for the subsequent works: to accurately modify the OS and IW with FE_{OS} and FE_{IW} , respectively. The thiol-modified DNAs served for ionic gating as FE_{IW} , and the hydrophobic molecules (or electric charged molecules) served for blocking interference molecules as FE_{OS} , respectively (as shown in Figure 4 and 5). Moreover, the fluorescent depth and intensity of interference characterized by LSCM are two key parameters to evaluate the anti-interference ability of FE_{OS} , which also the first application.

Finally, we are puzzling that the Reviewer 3# agrees with that the objective is different from our previous papers, which is also affirmed by Reviewer 1# and Reviewer 2#, but He/She still rejected our paper. The objective is the emphasis of one paper, which brings about the highlights and innovation of paper. Even if the content is similar, one paper

deserves to be recommended due to its innovation and highlights. Not to mention, the paper with apparently different contents.